# Molecular Characterization and Bioactivities of a Novel Polysaccharide from *Phyllostachys pracecox* Bamboo Shoot Residues

**DOI:** 10.3390/foods12091758

**Published:** 2023-04-23

**Authors:** Xubo Huang, Yalan Zhang, Na Xie, Junwen Cheng, Yanbin Wang, Shaofei Yuan, Qin Li, Rui Shi, Liang He, Min Chen

**Affiliations:** 1The Key Laboratory of Biochemical Utilization of Zhejiang Province, Zhejiang Academy of Forestry, Hangzhou 310023, China; 2Department of Food Science and Technology, College of Light Industry and Food Engineering, Nanjing Forestry University, Nanjing 210037, China; 3Bamboo Shoots Engineering Research Center of the State Forestry Bureau, Department of Bamboo, Zhejiang Academy of Forestry, Hangzhou 310023, China; 4Zhejiang Longyou International Trade Bamboo Shoots Co., Ltd., Quzhou 324400, China

**Keywords:** *Phyllostachys pracecox* bamboo shoot, valorization, polysaccharide, chain conformation, structure–function, immunomodulation

## Abstract

Dietary carbohydrates are unexploited in the by-products of economically valuable *Phyllostachys pracecox* bamboo shoots. A residue-derived polysaccharide (PBSR1) was aqueously extracted from the processing waste of this bamboo shoot species. Its primary structure and advanced conformation were elucidated by a combined analysis of spectroscopy, chromatography, 2D nuclear magnetic resonance, laser light scattering and atomic microscopy. The results indicated PBSR1 was a triple-helix galactan consisting of →6)-β-D-Gal*p* and →3)-β-D-Gal*p* in linear with an 863 KD molecular weight (M_w_). The relationship between the radius of gyration (Rg) and intrinsic viscosity ([η]) on M_w_ were established as R_g_ = 1.95 × 10^−2^M_w_^0.52±0.03^ (nm) and [η] = 9.04 × 10^−1^M_w_^0.56±0.02^ (mL/g) for PBSR1 in saline solution at 25 °C, which indicated it adopted a triple-helix chain shape with a height of 1.60 ± 0.12 nm supported by a red shift of λ_max_ in Congo red analysis. The thermodynamic test (TG) displayed that it had excellent thermal stability for the food industry. Further, those unique structure features furnish PBSR1 on antioxidation with EC_50_ of 0.65 mg/mL on DPPH· and an ORAC value of 329.46 ± 12.1 μmol TE/g. It also possessed pronounced immunostimulation by up-regulating pro-inflammatory signals including NO, IL-6, TNF-α and IL-1β in murine cells. Our studies provided substantial data for the high-valued application of residues and a better understanding of the structure–function relationship of polysaccharide.

## 1. Introduction

Bamboo shoots emerged from the ground are an edible inner part of new culms of bamboo which is widely distributed in Asia, Africa and South America throughout the world. It has a great reputation in terms of vegetarian food due to its high nutritional value, non-pollution and attractive flavor [1]. As one of the promising species of the genus Phyllostachys Sieb, *Phyllostachys pracecox* bamboo shoot is welcome in southeastern China with a richness in dietary carbohydrates and proteins but a lowness in fat [2]. However, only tender tips of bamboo shoots are adopted as cuisine or functional food. With the fast blooming of the food processing industry, huge tons of bamboo shoot residues (BSRs) occupying over 70% of the whole parts are treated as waste and normally abandoned into the environment, resulting in a waste of considerable resources and environmental pollution [3]. As a matter of fact, those discarded BSRs still have undeveloped compositions including dietary carbohydrates, polyphenols, proteins and amino acids [4].

Among those ingredients, dietary fibers, especially polysaccharides from a variety of bamboo shoots, have drawn great attention for their vital roles in the health of living organisms [5,6]. They have been proven to perform antioxidation, immunomodulation, antitumor, antiaging, hypoglycemic, hypolipemic and even stimulation to a number of intestinal beneficial microbiota [7,8,9,10]. Five polysaccharides with different molar ratios isolated from *Chimonobambusa quadrangularis* bamboo shoot had potent abilities to inhibit probiotic bacterial species from growing and promote the yield of short-chain fatty acids [11]. A heteropolysaccharide from its processing by-products was one kind of 1→6-linked mannan exhibiting strong antioxidant activities in free radical scavenging as well [12]. A glucan obtained from *Phyllostachys edulis* shoots was reported to show significant inhibition effects on free radicals [13]. The polysaccharide from shoots of *Guadua chacoensis* was identified to be glucuronoarabinoxylan, which showed promise for the food industry [14]. Generally, biological activities of polysaccharides would be significantly affected by physicochemical properties including solubility, molecular size, branching degree and glycosidic linkages, as well as triple-helical tertiary conformation [15]. They normally adopt various chain conformations in solution to bind certain receptors during the execution of a biological function, which enjoys more in-depth research [16]. Concerning that, it is urgent to get involved in the unique solution properties, advanced structural characters and biofunctions, which will be in favor of understanding the relationship between its structure and function. Unfortunately, there is little information about the structures and biological activities of polysaccharides from *Phyllostachys pracecox* bamboo shoot residues, which limits their further utilization in the food industry. In order to alleviate the environmental disruption caused by those disposed bamboo shoot residues, converting the agro-industrial waste into products with high added economic value is of great concern and worth further effort. Therefore, the aim of this study is to exploit some beneficial substances from hot-water-extracted crude polysaccharides (PBSR) of *Phyllostachys pracecox* bamboo shoot residues. A novel biomacromolecule (PBSR1) was fractionated correspondingly by a series of column chromatography via DEAE-Sepharose and Sephacryl S200. Then, its physicochemical properties and the advanced chain conformation were evaluated by a series of advanced analytical techniques. Moreover, its antioxidant activity on free radicals and immunomodulatory activity on cytokine release were tested based on murine macrophage RAW264.7 cells as well. The goal of the work is to comprehensively utilize the discarded bamboo shoot residues and offer some fundamental information about the structure–function relationship of bioactive polysaccharide for future valorization as a source of healthy food or food additives.

## 2. Materials and Methods

### 2.1. Materials and Reagents

*Phyllostachys pracecox* bamboo shoot residues were collected from the processing workshop in Zhejiang Shengshi Biotechnology Co., Ltd. (Huzhou, China). Monosaccharide standards were ordered from Shanghai Aladdin Bio-Chem Technology Co., Ltd. (Shanghai, China). DEAE-Sepharose FF and Sephacryl S-200 were purchased from Pharmacia Co. (Uppsala, Sweden). Trolox, Congo red, DPPH and 2,7-dichlorofluorescent diacetate (DCFH-DA) were bought from Sigma-Aldrich (St. Louis, MO, USA). Lipopolysaccharide, polymyxin B (PMB), CCK-8 kit, IL-6, IL-1β and TNF-α ELISA kits were supplied from Nuoyang Biotechnology Company (Hangzhou, China). All other chemicals were at analytical grade and obtained from Sinopharm Chemical Reagent Co., Ltd. (Shanghai, China).

### 2.2. Extraction and Fractionation of PBSR

After being smashed through a No. 60 mesh, the powdered raw bamboo shoot residues of *Phyllostachys pracecox* were preserved in 80% ethanol for 6 h to separate some colored small molecules. Then, the dried residues were subjected to aqueous extraction (95 °C) for 2 h under a solid-to-liquid ratio of 1:30. The resulting solution was reduced in a vacuum and followed by precipitation to reach a final ethanol concentration of 75%. The protein in the collected precipitate was continuedly removed by Sevage reagent [17]. With a removal by centrifugation (10,600× *g*, 5 min), the final crude polysaccharide was produced by dialysis (MWCO 4000 Da) and lyophilization, named PBSR.

A total of 50 mg/mL PBSR was prepared by dissolution in distilled water and filtration with a membrane (0.45 µm, Millipore, Burlington, MA, USA). Then, it was applied onto a DEAE-Sepharose FF column (2.0 cm × 35 cm) and successively flushed by a series of saline solutions (0–1 M NaCl at a flow rate of 1.0 mL/min with 3 mL fraction collected). Each tube was detected by the phenol-sulfuric acid method and the 0.1 M sub-fraction was pooled and dialyzed for further purification [18]. The collected elution was then subjected to a Sephacryl S-200 column (1.5 cm × 60 cm) eluted by 0.1 M NaCl at a flow rate of 0.5 mL/min. The main purified polysaccharide named PBSR1 was finally obtained after desalting and lyophilization. Its protein content was evaluated by BCA assay. The flavonoid content was determined by aluminum chloride assay and the m-hydroxybiphenyl method was conducted to assess the uronic acid content [19].

### 2.3. Physicochemical Analysis

#### 2.3.1. Monosaccharide Composition

The monosaccharide components of PBSR1 were investigated by the PMP-derived HPLC method in our previous procedures [20]. A total of 5 mg of the sample was hydrolyzed by 4 M TFA under 121 °C for 5 h and its residue was removed by the addition of methanol. Then, the hydrolyzed sample and 10 standard sugars were reacted with 50 μL of 0.5 M PMP after dissolution in 75 μL of 0.3 M NaOH. Afterward, the same volume of 0.3 M HCl was added to neutralize the solution and filtered with a 0.22 μm membrane (Millipore, MA, USA) prior to the injection. The PMP-derived samples were tested by injecting them into an Eclipse XDB-C18 column (4.6 × 250 mm × 5 μm, Agilent, Santa Clara, CA, USA) for UV detection at 245 nm. The flow rate was maintained at 1.0 mL/min under 30 °C. A mixture of 0.05 M PBS (pH 6.9)-acetonitrile (83:17) was adopted as the mobile phase.

#### 2.3.2. Homogeneity and Molecular Weight

Its absolute molecular weight and homogeneity were determined by the SEC-MALLS system coupled with two detectors following [21], which had two combined columns (TSK G5000 PWXL and TSK G3000 PWXL, Tokyo, Japan), a multi-angle laser light scattering detector (DAWN HELEOS II, Santa Barbara, CA, USA) and a refractive index detector (RID-10A, Niigata Shi, Japan). The tested condition was as follows: the mobile phase was a 0.15 M NaNO_3_ solution, the sample concentration was 3 mg/mL, the flow rate was set as 0.6 mL/min and 0.138 mL/g was selected as the refractive index increment dn/dc. Data processing was analyzed by ASTRA 7.1.2 software (Wyatt Technology, Santa Barbara, CA, USA).

#### 2.3.3. UV and FT-IR Analysis

PBSR1 with 1 mg/mL solution was scanned by a UV-1900 spectrometer (Shimazu Co., Kyoto, Japan) under 200–400 nm. A pellet for FT-IR was composed of sample powder (2 mg) and KBr (150 mg), which was scanned in the range of 4000–400 cm^−1^ by a Nicolet iS50 FT-IR spectrometer (Thermo Fishier Corporation, Waltham, MA, USA).

#### 2.3.4. Methylation Analysis

The methylation process was conducted followed by the former reference [22]. A total of 3 mg of dried PBSR1 was dissolved in 1 mL of anhydrous dimethyl sulfoxide (DMSO) and methylated with 1 mL of CH_3_I, which was slowly added, lasting 8 h under N_2_ protection. Then, the solution was extracted with chloroform and the distilled water was evaporated. This operation was repeated until the absorption peak in the O-H region (3000–3500 cm^−1^) of the IR spectrum disappeared. Afterward, the resulting PBSR1 was continuedly reacted with TFA (4 mL, 2 mol/L) at 110 °C for 3 h followed by reduction with NaBH_4_ to produce the acetylated product. The analysis was performed by an Agilent 7890A/5975C instrument with an HP-5MS column (30 m × 0.25 mm × 0.5 mm) (Thermo Co., Austin, TX, USA). The temperature program was heated up from 150 to 250 °C at 6 °C/min, then kept at 250 °C for 15 min. The PMAAs of the samples were analyzed by m/z and expressed as a relative percentage of each component.

#### 2.3.5. 2D NMR Measurement

A total of 60 mg of PBSR1 was replaced with D_2_O three times and then finally dissolved in 0.5 mL D_2_O for the NMR experiment. ^1^H NMR, ^13^C NMR, ^1^H-^1^H correlation spectroscopy (COSY), total correlation spectroscopy (TOCSY), nuclear Overhauser effect spectroscopy (NOESY), heteronuclear single quantum correlation spectroscopy (HSQC) and heteronuclear multiple-bond correlation spectroscopy (HMBC) were recorded by using a 5 mm CYBBFO at an ultra-low temperature and a Bruker AVANCE 600 MHz spectrometer (Bruker Group, Fällanden, Switzerland) at 333.15 K [21].

### 2.4. Advanced Structure and Physical Property of PBSR1

#### 2.4.1. Chain Conformation Determination

The advanced chain conformation of PBSR1 was evaluated by the SEC-MALLS system combined with triple detectors, which had an additional differential viscometer (ViscoStarTM II, Wyatt Technology, Santa Barbara, CA, USA). Other operating conditions were the same as in Section 2.3.2.

#### 2.4.2. Atomic Force Microscopy (AFM)

Atomic force microscope (AFM) images of PBSR1 were recorded on a Park XE-70 AFM (Park Scientific Instruments, Suwon, Korea). After a stock solution of 1 mg/mL was prepared with Milli-Q water, a drop of 2 μL PBSR1 solution (5 μg/mL) was dropped onto the freshly cleaved mica substrate and dried in the air for 1.5 h. The freshly prepared sample was mounted on the AFM stage and imaged under atmosphere (25 °C, relative humidity of 40~50%) in tapping mode [22].

#### 2.4.3. Congo Red Analysis

A sample stock solution of 2 mg/mL and sodium hydroxide of 1.0 M were prepared with Milli-Q water after 2 h stirring. Then, a testing tube was added with 1 mL of PBSR1 solution, 2 mL of Congo red and a series of volumes of alkaline to get the final concentrations of 0–0.5 M. Each reaction took place in the dark for 1 h and the maximum absorption wavelength (λ_max_) between 400 and 700 nm was determined by a UV spectrophotometer [23].

#### 2.4.4. Thermal Analysis

The thermal properties of PBSR1, including thermogravimetry (TG), differential thermogravimetry (DTG) and differential scanning calorimetry (DSC), were determined through a thermogravimetric analyzer (STA449F3, Netzsch, Selb, Germany). A total of 5 mg of the dried sample was placed in a pan and heated from 25 °C to 800 °C at 10 °C/min, with nitrogen as the protective gas [24].

### 2.5. Antioxidant Activities

#### 2.5.1. DPPH Scavenging Activity

The DPPH scavenging capacity of PBSR1 was investigated using the reported method of Zhang et al. [25] with some modifications. Briefly, PBSR1 was prepared with distilled water to make a series of concentrations (0, 0.1, 0.2, 0.4, 0.6, 0.8, 1.0, 1.2 mg/mL). Then, a 1 mL sample was fed with 1 mL of 0.2 mM DPPH solution in ethanol. After a thorough vortex, the mixtures were darkly kept at 25 °C for 0.5 h and the absorbance of 517 nm was tested immediately. Ascorbic acid was used as a positive standard. The inhibition percentage of DPPH free radicals was calculated as the following equation:Scavenging rate (%) = (1 − A_s_/A_0_) × 100 (1)
where A_s_ was the absorbance of the sample and A_0_ was the blank control solution without the sample.

#### 2.5.2. Oxygen Radical Absorbing Capacity (ORAC)

The method of the oxygen radical absorbance capacity (ORAC) was adapted from [26] with minor modifications. AAPH (12.8 mM), Trolox and fluorescein (35 nM) were dissolved with 75 mM phosphate buffer (pH 7.4). The total reactive mixture was 200 μL, containing 40 μL fluorescein, 20 μL sample or 20 µL of Trolox (6.25, 12.5, 25, 40 and 50 µM) and 140 μL AAPH. The steps of fluorescence measurement were performed at 37 °C, with an excitation wavelength of 485 nm and emission wavelength of 528 nm. The plate was automatically shaken before the first reading, and the fluorescence was recorded every 2 min for 98 min until the fluorescence value was near to zero. All determinations were implemented three times. The ORAC value was expressed as Trolox equivalents (µmol Trolox/g).

### 2.6. Immunomodulatory Activity

#### 2.6.1. Cell Viability of RAW264.7

A CCK-8 kit was used to test the cell viability of RAW264.7 macrophages [21]. The murine cells were initially cultivated in 96-well plates for 24 h using PMB to avoid LPS contamination. Then, each well was replaced with 6.25–200 μg/mL PBSR1 or positive LPS (5 μg/mL) for another 24 h incubation. With the addition of CCK-8 for 37 °C maintenance, the value at 450 nm was tested immediately.

#### 2.6.2. Pinocytic Assay

The pinocytic activity of RAW264.7 cells was investigated by neutral red assay [27]. A total of 1 × 10^6^ of RAW264.7 cells were initially seeded in DMEM medium for 24 h and then added with PBSR1 (25–200 μg/mL) and LPS (10 μg/mL) for continual incubation. After 24 h, 0.1% neutral red was supplemented as an indicator followed by 100 μL of glacial acetic acid and ethanol. With a 2 h reaction, the absorbance at 540 nm was evaluated by a microreader.

#### 2.6.3. ROS Generation Measurement

2,7-dichlorofluorescent diacetate (DCFH-DA) was applied for ROS production according to the methods of [27]. RAW 264.7 cells were initially seeded into 96-well plates for 24 h. Next, the DMEM culture medium, PBSR1 samples (25–200 μg/mL) and LPS (10 μg/mL) were supplied into the wells for another 24 h, respectively. After the abandonment of all the culture solution, 100 μL of DCFH-DA (10 μM) was added into each well at 37 °C for 20 min dark incubation. Then, the cells were washed with PBS buffer and subjected to fluorescent intensity at 485 nm excitation and 538 nm emission.

#### 2.6.4. Cytokines Production

Four cytokine levels including NO, IL-6, TNF-α and IL-1β cytokines were measured according to [28]. Initially, macrophages were treated with PBSR1 (25–200 μg/mL) or LPS (10 μg/mL) for 24 h. Then, the supernatants were collected for the level determination of NO, IL-6, TNF-α and IL-1β following a commercial ELISA kit’s instructions.

### 2.7. Statistical Analysis

All the analyses on antioxidant and immunomodulatory activities were performed in triplicate. The data were recorded as mean ± standard deviation. Student’s *t*-tests were used to analyze the statistical significance of differences. Differences with *p* < 0.05 were considered significant and differences with *p* < 0.01 were considered extremely significant.

## 3. Results

### 3.1. Extraction and Fractionation of PBSR

After a series of extraction processes including hot-water diffusion, ethanol precipitation and protein removal, the crude PBSR was obtained from the bamboo shoot residues of *Phyllostachys pracecox* with a yield of 2.24%. Then, the PBSR was isolated by a DEAE-Sepharose FF column with gradient saline elution to present the two elution peaks in Figure 1A: PBSR1 (87.6%) and PBSR2 (12.4%), tested by the phenol-sulfuric acid method. It was obvious that the first fraction eluted by 0.1 M saline accounted for the majority of the crude polysaccharides in the bamboo shoot residues of *Phyllostachys pracecox.* Therefore, PBSR1 was continually subjected to be purified on Sephacryl S200, which is presented as a symmetrical sharp peak in Figure 1B. It was notable that there was no protein content in PBSR1 using the BCA assay and no uronic acid was detected at 525 nm. The flavonoid content was 1.67% in the dry extract, while the carbohydrate content occupied almost 97.38% in the dry sample [29]. Zheng et al. reported that there were 1.52% crude polysaccharides from the bamboo shoot (*Leleba oldhami* Nakal) shells, which was much lower than PBSR [30]. However, the yield (9.96%) of polysaccharides from the processing by-products of *Chimonabambusa quadrangularis* was higher than that of PBSR [12]. Concerning the differences in preparation, this reason might be explained by the fact that there were some monosugars and oligosaccharides inside the *Chimonabambusa quadrangularis* polysaccharides due to no precipitation on the extracted solution.

### 3.2. Monosaccharide Compositions of PBSR1

The variation of monosaccharide components may account for its special biological activity, which can be analyzed after PMP derivation by HPLC. Figure 2 shows the profiles of 10 standard sugars and PMP-derived PBSR1. It was quite notable that there was almost only one big peak of galactose in this fraction, suggesting it was one kind of galactan with negligible amount of arabinose inside (Gal/Ara ratio was 18:1). This differed from the findings of other reported AG-II and other types of bamboo shoots. Normally, AG-II is one of the hemicelluloses in natural plants, in which the arabinose content occupies around 20% of the total sugar, while that value in PBSR1 was less than 1%, suggesting galactose was the only major component in this biopolymer [31]. BSSP2a obtained from bamboo shoot (*Leleba oldhami* Nakal) shells was determined to have arabinose, xylose, mannose, glucose and galactose with a molar ratio of 20.4:4.9:1:3.4:20.6 [30]. The polysaccharides obtained from bamboo shoot by-products of *Chimonobambusa quadrangularis* mainly contained glucose, galactose and arabinose [32]. The polysaccharides extracted from the shoots of *Phyllostachys edulis* (Carr.) consisted of two kinds of monosaccharides, including xylose and arabinose [17]. A new β-glucan was isolated from the bamboo shoots of *Phyllostachys edulis* with a strong anticomplementary [10]. The purified PBSR1 was consistent with the results of PBSS2 from *Phyllostachys heterocycle* bamboo shoot shells, which was a kind of arabinogalactan [20].

### 3.3. Homogeneity and Molecular Weight of PBSR1

MW can be an inherent property of biomacromolecules and is closely related to their bioactivities. As shown in Figure 3A, the purity of PBSR1 had been proven by the single symmetrical peak in the SEC spectrum, which was detected simultaneously by three signals including laser light (LS), refractive index (dRI) and differential pressure (DP). The result evidenced the good performance of the purification process by Sephacryl S-200. Its molecular parameters containing the absolute molecular weight (M_w_), the number-average molecular weight (M_n_), the radius of gyration (R_g_) and the hydrodynamic radius (R_h_) were 1.15 × 10^5^ g/mol, 8.63 × 10^5^ g/mol, 23.0 nm and 20.7 nm, respectively. Moreover, 1.33 of the polydispersity (PDI) reflected a relatively narrow distribution of PBSR1 chains in saline solution in Figure 3B, which confirmed the homogeneity by gel purification [20]. Compared with other arabinogalactans, PBSR1 significantly differs from other Siberian larch AG (a weight average molecular mass of 9000–13,000) by its relatively higher molecular mass, even though its PDI was much lower than this kind of Siberian larch AG (PDI:1.9–2.3) [33]. Considering the properties of a lower R_h_ (9.6 nm) and highly branched conformation in the solution, PBSR1 may have an extended linear structure.

### 3.4. UV and FT-IR Analysis

There was a smoothly decreasing curve from 200 to 400 nm with no peaks at either 260 nm or 280 nm in Figure 4A, indicating PBSR1 did not contain uronic acid and protein. The FTIR spectrum presented a classical property of the polysaccharide functional group of PBSR1 in Figure 4B. The strong band at 3388 cm^−1^ was ascribed to the O-H stretching vibration. The peak at 2926 cm^−1^ was caused by the C-H stretching vibration. The C-O stretching vibration and the C-H angular vibration can be found by two adsorption peaks of 1479 cm^−1^ and 1245 cm^−1^. The obvious adsorption signal at 1084 cm^−1^ was attributed to the glycosidic linkage C-O-C stretching vibration. Similar peaks were found in the results of polysaccharides from *Chimonobambusa quadrangularis* processing by-products [32]. Two characteristic signals at 867 cm^−1^ and 782 cm^−1^ explained that PBSR1 may contain β-glycosidic bonds [30]. Both spectroscopic analyses further depicted that PBSR1 was one kind of β-linked galactan with no acid or protein inside, which was consistent with the former purity identification and component analysis.

### 3.5. Methylation Analysis

The information on glycosyl linkages in PBSR1 was achieved by methylation analysis. After methylated derivatization, the partially methylated alditol acetate (PMAA) derived from the sample yielded two major fragmentation patterns in Figure 5, a 2,3,4-tri-O-methyl-galactose and a 2,4,6-tri-O-methyl-galactose with the relative molar proportion of 14.3: 85.7, which reinforced two types of linkages in the residues, namely, 1,6-linked Gal*p* and 1,3-linked Gal*p* (Table 1). With regard to each percentage of the molar ratio, it suggested PBSR1 was a backbone of (1→6)-linked D-Gal*p*. Normally, type I arabinogalactans have a linear (1→4)-β-D-Gal*p* backbone with L-Ara*f* substituted at C-3, while type II arabinogalactans consist of predominantly (1→3)- but also (1→6)-linked β-D-Gal*p* chains [31]. So, our study was consistent with LP100R obtained from peach gum, which was identified to have a β-D-(1→6)-galactan backbone and was branched at O-3 [34]. From this point, PBSR1 belongs to type II arabinogalactans. However, taking the glycosidic linkages into account, PBSR1 was quite distinct from the traditional larch AGs-II. Most AGs-II are constituted of 1→3-linked β-D-Gal*p* as the backbone branched by 1→6-linked β-D-Gal*p* residues. As a special AG-II, PBSR1 has the opposite types of glycosidic bonds. This result may explain the differences in homogeneity and molecular weight compared to other larch AGs-II [35,36].

### 3.6. NMR Analysis of PBSR1

In order to clearly elucidate the fine structure of PBSR1, one-dimensional (^1^H NMR, ^13^C NMR) and two-dimensional (COSY, HSQC, HMBC NOESY) NMR techniques were employed to detect the glycosidic sequences of all the fragments. Two major anomeric proton signals were displayed at δ 4.46 and δ 4.71 ppm in ^1^H NMR (Figure 6A), suggesting both two residues contained β-configuration glycosidic bonds due to the chemical shifts below δ 5.0 ppm [27]. Residues A and B were annotated accordingly for the facilitation of the pattern determination, which was confirmed by two significant intercorrelated signals in the anomeric region of the HSQC spectrum (Figure 6D). Other proton signals (H2–H6) were found in the region of δ 3.51–4.35. ^13^C NMR provided the corresponding anomeric carbon resonances at δ 103.31 and δ 103.78 ppm (Figure 6B). All the chemical shifts of the proton for each residue could be assigned by the COSY spectrum in Figure 6C. Followingly, their corresponding carbon resonances were dug out in ^13^C NMR and HSQC spectra. Those chemical shifts are summarized in detail in Table 2.

For Residue A, the other proton resonances (from H-2 to H-6) were deduced from the correlation peaks in the COSY and NOESY spectra due to their responses in the same spin system. The HSQC spectrum revealed their related carbon/proton chemical signals at δ 70.73/3.55, δ 72.64/3.66, δ 68.62/3.93, δ 75.10/3.72 and δ 69.01/3.93(4.06) ppm, respectively. The obvious downfield shift of C-6 suggested it was β-1,6-D-galactopyranose, which agreed with the results of β-(1→6)-linked arabinogalactan from *Mangifera indica* L. fruit exudate [37]. Likewise, the proton and carbon chemical shifts of Residue B were assigned from the correlation peaks in the COSY along with the HSQC spectrum. Arising from the downfield resonance of the C-3 group at δ 81.38 ppm in the HSQC spectrum, Residue B was deduced to be β-1,3-D-galactopyranose. Concerning the strong intense anomeric proton signal at δ 4.46 ppm, it further proved that Residue A was the predominant glycosidic linkage in PBSR1, which was consistent with the former methylation results.

The sequence of two residues was elaborated by the HMBC and NOESY spectra. The strong inter-cross signals at δ 4.46/103.31 and δ 3.66/103.31 ppm in the HMBC spectrum implied that the backbone chain of the polymer mainly consists of (1→6)-D-galactopyranan. The inter-residual cross-peaks from A H-1/B C-3 (δ 4.46/81.38 ppm) to A C-1/B H-3 (δ 103.31/3.89 ppm), A H-6/B C-1 (δ 4.06/103.78 ppm) and A C-6/B H-1 (δ 69.01/4.71 ppm) speculated that Residue B was linearly linked to Residue A in the major chain. That deduction could be reinforced by the NOE correlation peaks at A H-1 (δ 4.46 ppm) to B H-3 (δ 3.89 ppm) and A H-6 (δ 4.06 ppm) to B H-1 (δ 4.71 ppm) in Figure 6F. Consequently, PBSR1 was one kind of galactan consisting of →6)-β-D-Gal*p* and →3)-β-D-Gal*p* in linear. Its repeating units were established in Figure 6G. Most of the type II arabinogalactans obtained from *Larix laricina* were determined to have a 1,3-linked Gal*p* backbone attached at C-6 to 1,6-linked Gal*p* side residues [38]. Taking the proposed structure, PBSR1 was slightly different from the traditional type II arabinogalactans, in which →6)-β-D-Gal*p* was branchly attached to the main chain of →3)-β-D-Gal*p* [31], while it was similar to the cell wall galactans from Flax (*Linum usitatissimum* L.), which could be acquired by the collection of flax suspension-cultured cells [39]. The results confirmed the analysis of monosaccharide composition, FT-IR and methylation.

### 3.7. Advanced Conformation and Physical Property of PBSR1

#### 3.7.1. Chain Conformation

The specific conformational properties could endow biomacromolecules with particular biological features, which have attracted great attention. In this study, the advanced chain parameters of PBSR1 were investigated in order to clearly find its structure–function relationship by a combination of static and hydrodynamic scattering techniques. The fractal dimension (d_f_) has now been widely used in the nature of structural insights for hyperbranched polysaccharides by the establishment of the relationship between R_g_ and M_w_. Generally, the d_f_ value of 1 means a rigid rod for polymer. A Gaussian coil-like polymer can be indicated by the d_f_ value ranging from 5/3 to 2 [40]. Above that d_f_ value, an object has a homogeneous density or dendrimers. It is defined as the inverse of the exponent ν, which can be extracted from the equation R_g_ = kM_w_^1/df^. The plot of R_g_ versus M_w_ for PBSR1 was presented in Figure 7A and its resulting relationship was described as R_g_ = 1.95 × 10^−2^M_w_^0.52±0.03^ (nm). The calculated d_f_ value was to be 1.92, speculating that the chain shape of this macromolecule adapts a random coil in saline. Liu et al. [20] found that the value of d_f_ for PBSS was 2.52, which explained the existence of a compacted structure in the solution. Some dendrimers with a d_f_ value of 3 were reported by other researchers [41].

Another function has also been applied to explore the relationship between intrinsic viscosity ([η]) and the molecular weight (M_w_) regarding the architecture of polysaccharide in solution. By utilizing the Mark –Houwink equation ([η] = KM^α^), the crucial conformational factor, exponent α could be experimentally determined. In addition, this exponent is usually an indicator of the feature of biopolymer chains in a solvent. In the light of polymer theory, the α value of 0.33 reflects the sphere-like structure of a polymer molecule [41]. A value between 0.5 and 0.8 indicates a flexible chain and a larger value than one for a stiff rod of macromolecule. Figure 7B shows the plot of [η] dependences on the M_w_ for PBSR1 0.15 M NaNO_3_ at 25 °C and the Mark–Houwink equation for PBSR1 was expressed as [η] = 9.04 × 10^−1^Mw^0.56±0.02^ (mL/g). The slope (α = 0.56) suggested PBSR1 adopted a coil-like chain shape in saline, which was in good accordance with the analysis of the R_g_-M_w_ function. However, the chain conformation was quite different from the results of arabinogalactan from bamboo shoot shells of *Phyllostachys heterocycla* [20] and β-D-glucan from sclerotia of *Pleurotus tuber-regium* [42], both of which existed as a compacted sphere chain in the solution. Considering the glycosidic linkages of PBSR1, those differences in conformation might be ascribed to little inter-molecular hydrogen bonds in PBSR1 compared with the other two structures. From this point, the specific monosaccharide linkages may significantly affect its advanced architecture.

#### 3.7.2. AFM of PBSR1

AFM has been effectively employed to image individual macromolecules directly in an air or solvent environment at a nanometer scale. After air-depositing on the mica of 10 μg/mL PBSR1, its AFM images were presented in Figure 8. The results proved that the molecules appeared to be composed of kinked and segmented stands. An overwhelming percentage of those tended to be aggregated end to end or side to side, which might be caused by the drying process. Moreover, some of the stands curled to form closed loops. On average, these chains had a height of 1.60 ± 0.12 nm and a width of 157 ± 0.23 nm. From the value of the height, single PBSR1 molecules might adopt a helical conformation on the substrate, which agreed with the finding of xyloglucan from *Guibourtia hymenifolia* Leonard seeds [43]. Similar chain structures have been found in other bamboo shoots resources. A β-pyran polysaccharide from bamboo shoot shells had also been observed to form a stratified and closely arranged structure [30]. A 1,4-linked glucan from bamboo (*Phyllostachys edulis*) shoot presented a flexible chain with fewer branches and its uniformed height was estimated to be 0.25–1.5 nm. Our AFM observation further proved the analysis of NMR and light scattering, which was a kind of linear and flexible galactan in the solution.

#### 3.7.3. Congo Red Analysis

The Congo red method can easily find the triple-helix configuration of polysaccharides by binding them to form a complex with a slight red shift of λ_max_ in NaOH solvent. Figure 9A described the λ_max_ curves of Congo red and Congo red-PBSR1 complex in a series of NaOH concentrations from 0–0.5 M. It was found that the λ_max_ of the sample complex had an obvious bathochromic shift at 0.1 M alkaline solution, indicating this sugar contained a triple helical structure in the diluted NaOH environment, which was in great accordance with the observation by AFM and analysis of light scattering. A hetero-polysaccharide bamboo from Chimonobambusa quadrangularis shoots by-products had also been reported to have a similar helix structure [12].

#### 3.7.4. TG Analysis

The importance of the thermodynamic properties of polysaccharides, which are cruicial for food processing due to their thermostability and are significantly affected by their own chemical structure, is common awareness. The information of thermal weightlessness and endo/exothermic changes can be obtained by analyzing TG and DSC patterns. From the TG and DTG profiles in Figure 9B, there were three stages of weight loss in PBSR1. A total of 14.5% of weight loss was shown in the initial stage (25–100 °C), which was explained by the loss of adsorbed water in PBSR1. There was no significant weight change when the temperature was increased from 100 °C to 250 °C, reflecting that the sample was relatively stable below 250 °C. Then, the temperature continually climbed up to 500 °C, and the severe cleavage of the glycosidic groups caused a weight loss of 78.2%. In addition, the maximum decomposition rate of PBSR1 appeared at 320.2 °C from the analysis of the DTG curve. The last stage happened between 500 and 850 °C, with only 15.5% of residue left to produce ash. The DSC analysis in Figure 9B suggested that PBSR1 had both endothermic and exothermic reactions. The first peak indicated the appearance of an endothermic reaction at 102.2 °C, ascribed to the evaporation of bound water in the sample component. The other exothermic reaction occurred at 333.2 °C, explained by the thermal decomposition of the sample. Compared with the degraded points of other bamboo shoot polysaccharides [32,44], the galactan isolated from these Phyllostachys pracecox bamboo shoot residues manifested better thermal stability and promising application in the food industry.

### 3.8. Antioxidant Activities In Vitro

#### 3.8.1. DPPH Scavenging Ability

The existence of free radicals in the human body is responsible for oxidative injury and other diseases, which can be eliminated by natural biomacromolecules. In this study, a DPPH scavenging assay was employed to test the inhibitory effect of PBSR1 compared with Vc, and the result is presented in Figure 10A. The purified PBSR1 exhibited a preferable scavenging ability on the DPPH radical in a dose-dependent pattern, although its ability was not as remarkable as that of Vc. Initially, the free radicals’ removal from the sample could be negligible in the low concentration. Then, there was an increase in its scavenging effect at the concentration ranging from 0.4 to 1.0 mg/mL, beyond which the capacity gently climbed up to 89.2% at 1.2 mg/mL. The EC_50_ value of PBSR1 was tested to be 0.65 mg/mL, which was extremely lower than those of different pretreated polysaccharides from Chimonobambusa quadrangularis processing by-products, in which the EC_50_ values of freeze-drying CPS, vacuum-drying CPS, hot-air drying CPS and spry-drying CPS were 1.31 mg/mL, 1.51 mg/mL, 2.17 mg/mL and 2.53 mg/mL, respectively [45]. This means that PBSR1 from Phyllostachys pracecox bamboo shoot residues could be accepted as a better antioxidant than those from Chimonobambusa quadrangularis resources.

#### 3.8.2. ORAC Capacity

The ORAC is a convincing method of evaluating the antioxidant capability of different foods, which can be acquired by the analysis of the kinetic protective behavior of samples against fluorescent decay induced by AAPH. As seen in Figure 10B, the fluorescent decay curves of Trolox solutions and the tested sample with a concentration from 5 μM to 50 μM were above that of the blank. There was a good linear relationship between the net area under the curve (AUC) and the standard Trolox, which was shown by the linear curve in the top right-hand corner (y = 0.8415x + 4.0264, R^2^ = 0.9887). Based on that equation, the ORAC value of PBSR1 was analyzed to be 329.46 *±* 12.1 μmol TE/g, remarkedly higher than those of treated polysaccharides from other different species [12,17,30]. Among them, the strongest ORAC of the freeze-drying CPS was only 197.79 *±* 22.5 μmol TE/g. The reason might be attributed to the difference in purity and structural features. The stretched PBSR1 molecules might make hydroxyl groups on the chain exposed to the environment, easily providing hydrogen to stabilize free radicals or directly react with them to terminate the chemical reaction.

### 3.9. Immunomodulatory Activity of PBSR1 on RAW 264.7 Cells

#### 3.9.1. Cell Viability

It is essential to test the cytotoxicity of drugs on RAW 264.7 cell lines prior to the investigation of immunomodulatory activity. Figure 11A depicted the results of the cell viability of PBSR1 on RAW 264.7 cells by using the CCK-8 method. Within the tested range of 25–400 μg/mL, the cell viabilities of PBSR1 were 100.5%, 102.4%, 106.6%, 112.1% and 114.6%, respectively. It was obvious that this sample exerted no toxicity on the immune cells and had proliferation to some extent on that.

#### 3.9.2. Pinocytotic Activity

The inner immune system of the human body can be triggered to play pinocytic and phagocytic functions after the sense of foreign invasion by an unknown microorganism. A neutral red uptake assay is an effective and simply practicable tool to garner that information by measuring the production of neutral red substances. After 24 h treatment by PBSR1, the pinocytic rates presented an up-forward trend with an increasing concentration of 25–200 μg/mL (Figure 11B). At the concentration of 200 μg/mL, the maximum cell uptake intensity of PBSR1-treated cells was 172.3%, which was quite close to 89.5% of the LPS-treated group (*p* < 0.05). The results indicated that PBSR1 could enhance the pinocytic activity significantly. Similar structural polysaccharides have been reported by Yang et al. [46] to possess strong immune activities, where a linear α-1,6-galactan had stimulated the pinocytosis and phagocytosis of macrophages. So did some AGs, as natural immunostimulators, they have been evidenced to specifically act on antigenic immune response as vaccines [31].

#### 3.9.3. ROS Analysis

ROS, regarded as the second signaling messenger, has been evidenced to be correlated with the synthesis and secretion of inflammatory cytokines and chemokines [47]. It can promote pinocytic activity of macrophage by the formation of macropinosome via the enhancement of respiration. In this study, the ROS generation of RAW264.7 cells treated with certain concentrations of the sample was tested by detecting DLF fluorescent intensity. As shown in Figure 11C, PBSR1 gave a dose-dependent increase of fluorescent intensity of the treated murine cells in the range of 25–200 μg/mL compared to the control group. The value of ROS produced by PBSR1 at a concentration of 200 μg/mL could climb to 1.23, which was 90.4% of LPS at 10 μg/mL. The results presented show that PBSR1 could upregulate the production of intracellular ROS of RAW264.7 cells.

#### 3.9.4. Cytokines Production and the Relationship of Structure–Function

It is considered that macrophages can defend the host against foreign infections and external pathogens by releasing chemokines and cytokines including NO, IL-6, TNF-α and IL-1β, etc. Among them, NO plays the role of the killer to eliminate parasites and tumor cells. Other cytokines are closely associated with the signal transduction of inner cell immunomodulation by the participation of proliferation, differentiation and receptor expression [48]. Figure 11D–G showed the secretion of four cytokine representatives treated by the PBSR1 sample with different concentrations. As expected, PBSR1 stimulated dramatically the cell secretion of NO, IL-6, TNF-α and IL-1β in a clear concentration-dependent manner. Specifically, the secreted levels of those four inflammatory factors were tested to be 18.3 μmol, 1863.9 pg/mL, 903 pg/mL and 35.8 pg/mL after 24 h treatment of the 100 μg/mL sample. Moreover, the content of IL-1β could reach 43.7 μmol after the exposure of cells to 200 μg/mL PBSR1, almost as high as the value of the LPS group at 10 μg/mL (*p* < 0.01). A similar phenomenon can be found in the secretion of the other three cytokines and other polysaccharides with structural proximity. At the same level, 3-O methylated galactan with 1.8 × 10^4^ g/mol isolated from *Cantharellus cibarius* promoted the secretion of IL-6, TNF-α and IL-1β [46]. The immunomodulatory capacity of AG (2.07 × 10^5^ g/mol) extracted from *Carthamus tinctorius* caused a strong stimulation effect on RAW 264.7 macrophages and NO-production [49].

There is a consensus that molecular weight, monosaccharide components, molecular size, glycosidic types and chain conformation, either alone or simultaneously, affect the overall immunostimulation of polysaccharide [40]. The reason why PBSR1 exerted a potent immunomodulatory effect on RAW 264.7 cells is related to its intrinsic structural features and could possibly arise from its moderate M_w_, β-(1→6)-galactopyranosyl backbone and triple-helix conformation. Although AGs with a lower MW and less compacted structure presented higher immunomodulatory effects than other AGs acquired from *Boswellia carterii* [48] and *Saposhnikovia divaricata* [50], some contradictions have revealed that the galactan with a higher M_w_ (4.03 × 10^6^ g/mol) from *Moringa oleifera* leaves possessed a superior effect on the immune system [51]. Our purified PBSR1 has a relatively high M_w_ compared with most reported galactans. Moreover, some studies suggested that the side chains of β-(1→6)-galactaoligosaccharides in Astragalus mongholics polysaccharide played a crucial role in intestinal Peyer’s patch-immunomodulation. The enzymatic-degraded deletion of β-(1→6)-galalactans side branches from the main chain caused significant weakness in the immune effect [38]. With this point in view, the repeating units of β-(1→6)-galactaoligosaccharides might be crucial for their biological function as active binding sites [31]. Further, triple helical conformation is regarded as a crucial structure characteristic for biomacromolecule’s immune activity. It has been evidenced that Letinan’s immune or antitumor activities were lowered dramatically when its tertiary structure was broken [52]. The same was true of schizophyllan [53]. Considering those, the existence of a helical structure formed by hydrogen bonds in galactan might have contributed to the bioactivity and prevailed over the hindrance imposed by the larger size or higher M_w_ of polysaccharide. Consequently, our results revealed that PBSR1 with the unique conformation manifested pronounced immunostimulatory activities by up-regulating some key pro-inflammatory signals such as NO, IL-6, TNF-α and IL-1β in murine cells.

After further deep investigation of the structure–function relationship, the immunological properties of polysaccharide can be ascribed to its unique structure features including a moderate MW, special backbone, functional groups and specific conformation. Besides the classical type II AGs, other galactans obtained from marine organisms such as carrageenans and porphyrans have been reported to have immunostimulatory activity. The former have proved that a number of sulfate groups conferred its strong bioactivity [54] and the latter have a linear backbone consisting of 3-linked β-D galactosyl units, which was conducive to its potent activity [55]. With respect to the advanced architecture, a triple-helix conformation is usually more stable than the single chain conformation due to the participation of H bonds in the intermolecular chains. β-glucan has been evidenced to possess satisfactory immune activity because its triple-helix conformation gifted a highly ordered feature, boosting the recognition by the immune receptor (Dectin-1) [56]. Concerning all that above, our data led us to speculate that the special structure properties with its moderate M_w_, the repeating units of β-1,6-linked Gal*p* and unique triple-helix architecture conferred PBSR1 to the strong immunostimulatory activities. All those findings boosted PBSR1 to be developed as a food additive or healthy food in functional food, cosmetics and pharmaceutical industries.

## 4. Discussion

There are a number of dietary carbohydrates that are still left in the by-products of economically valuable *Phyllostachys pracecox* bamboo shoot during food processing. In this study, a novel natural polysaccharide (PBSR1) was isolated and purified from this bamboo shoot waste. Its glycosidic linkages and advanced chain structure were investigated by HPLC, FT-IR, methylation, 2D-NMR, MALS and AFM. The results showed PBSR1 was a 1→6-β-D-galactan with 863 KD M_w_. The function of R_g_ versus M_w_ and the Mark –Houwink equation for PBSR1were found as the following: R_g_ = 1.95 × 10^−2^M_w_^0.52±0.03^ (nm), [η] = 9.04 × 10^−1^M_w_^0.56±0.02^ (mL/g) in 0.15 M NaNO_3_ solution at 25 °C, which illustrated it adopted a random coil chain with a height of 1.60 ± 0.12 nm on average supported by an obvious red shift of λ_max_ in alkaline solution by Congo red assay. The TG analysis proved it can be developed in the food industry with excellent thermal stability. Moreover, PBSR1 had better antioxidant scavenging on DPPH· with EC_50_ of 0.65 mg/mL and its ORAC value was to be 329.46 ± 12.1 μmol TE/g compared to other polysaccharides from *Chimonobambusa quadrangularis* processing by-products. It manifested great immunostimulated activity by up-regulating pro-inflammatory signals including NO, IL-6, TNF-α and IL-1β in RAW264.7 cells. This specific architecture with a moderate M_w_, β-(1→6)-galactopyranosyl backbone and triple-helix conformation may be crucial for its own bioactivities. Our data provided the promising possibility of deep utilization of the wastes and a better understanding of the structure–function relationship of polysaccharide.

## Figures and Tables

**Figure 1 foods-12-01758-f001:**
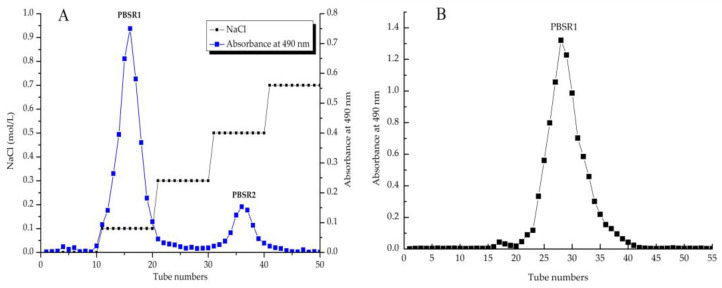
The elution curve of PBSR from the bamboo shoot residues of *Phyllostachys pracecox* on DEAE−Sephyrose FF column (**A**), and the purified profile of PBSR1 eluted by Sephycryl S200 column (**B**).

**Figure 2 foods-12-01758-f002:**
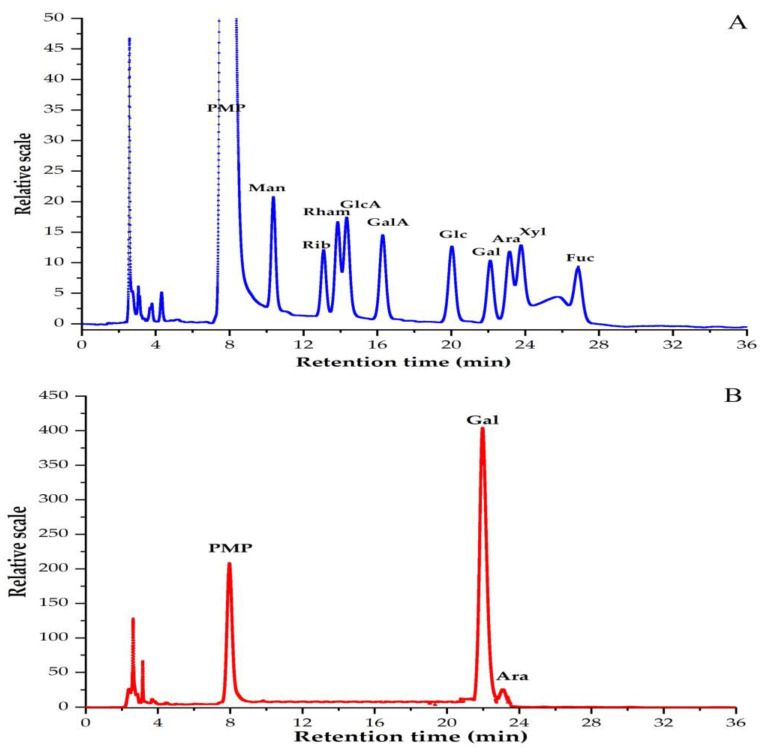
HPLC chromatograms of PMP derivatives of standard monosaccharides (**A**) and PBSR1 (**B**) detected at 245 nm with a flow rate of 1 mL/min.

**Figure 3 foods-12-01758-f003:**
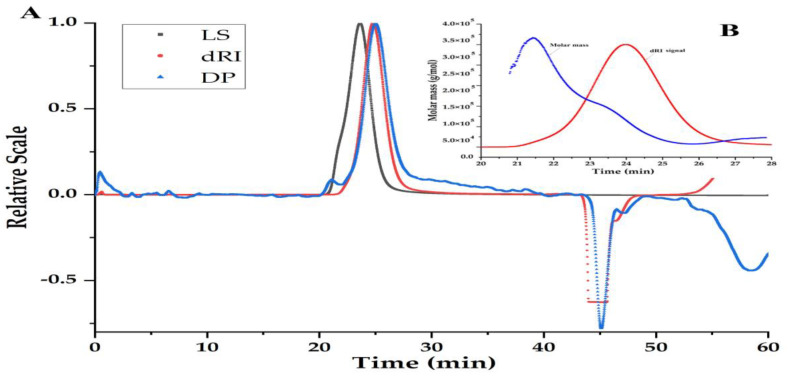
Size exclusion profiles of PBSR1 in 0.15 M NaNO_3_ solution at room temperature detected by three signals including laser light (LS), refractive index (dRI) and differential pressure (DP) (**A**); molar mass profile of SEC−MALLS chromatogram for PBSR1 (**B**).

**Figure 4 foods-12-01758-f004:**
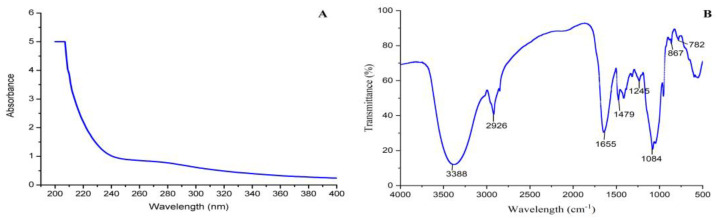
UV spectrum of PBSR1 scanned from 200−400 nm (**A**); FT−IR spectra of PBSR1 detected in the range of 4000−400 cm^−1^ at 4 cm^−1^ resolution (**B**).

**Figure 5 foods-12-01758-f005:**
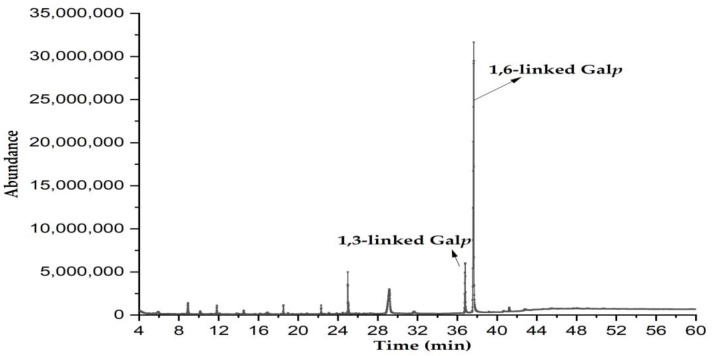
The total ion chromatograms of methylated PBSR1 operated from 150–250 °C.

**Figure 6 foods-12-01758-f006:**
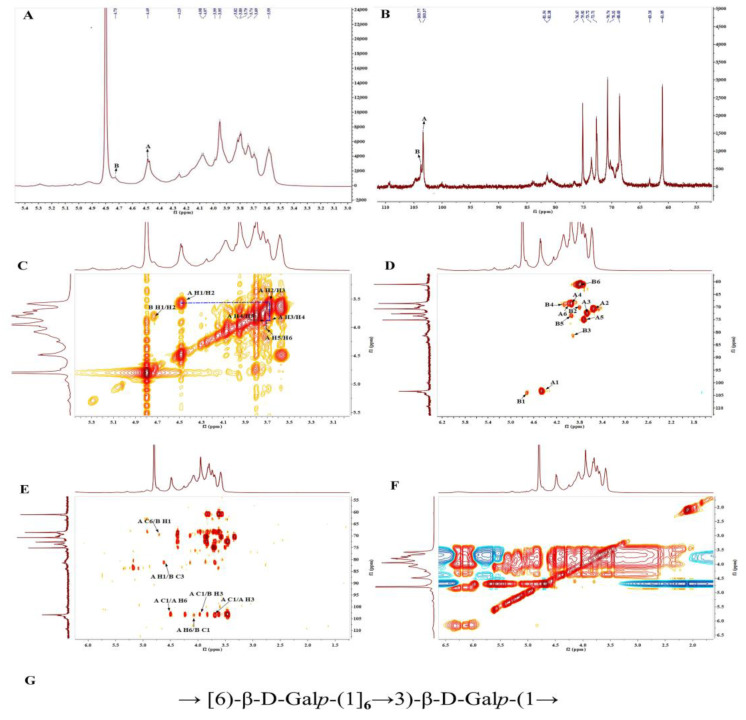
The NMR spectra of PBSR1. (**A**) ^1^H NMR spectrum, (**B**) ^13^C NMR spectrum, (**C**) ^1^H/^1^H COSY spectrum, (**D**) HSQC spectrum, (**E**) HMBC spectrum, (**F**) NOESY spectrum, (**G**) proposed structure of PBSR1.

**Figure 7 foods-12-01758-f007:**
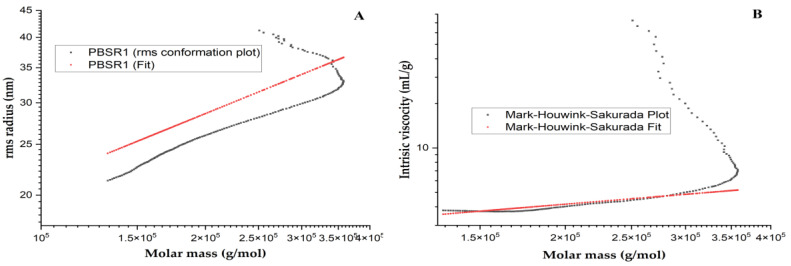
The plot of log R_g_ versus log M_w_ for PBSR1 in 0.15 M NaNO_3_ at 25 °C (**A**); [η] dependences on the M_w_ for PBSR1 in 0.15 M NaNO_3_ at 25 °C by applying the Mark–Houwink–Sakurada equation (**B**).

**Figure 8 foods-12-01758-f008:**
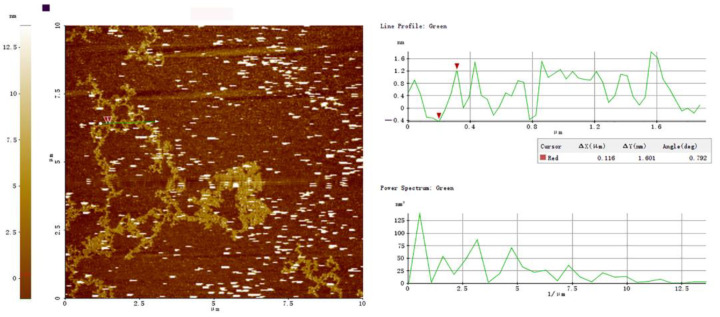
Tapping-mode AFM images of PBSR1 at 10 μg/mL on mica with scan size of 10 μm × 10 μm. The green cross-sectional profile was shown on the right of each AFM image.

**Figure 9 foods-12-01758-f009:**
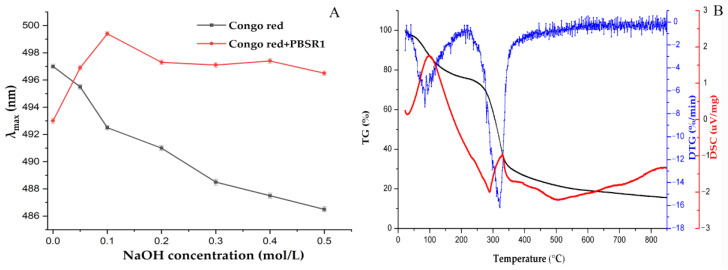
Maximum absorption wavelength of Congo red-PBSR1 complex at various concentrations of NaOH (**A**) and the thermal properties of TG analysis patterns and DSC curves for PBSR1 (**B**).

**Figure 10 foods-12-01758-f010:**
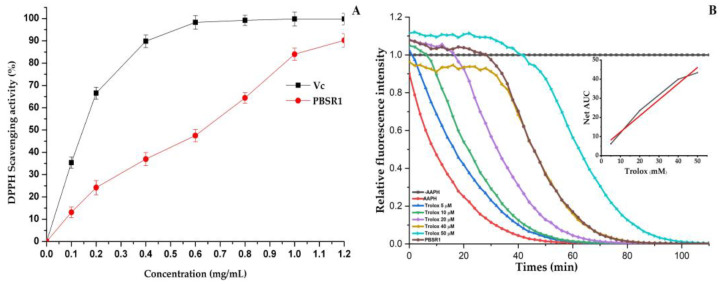
Antioxidant activities of PBSR1 in vitro. (**A**) DPPH radical scavenging activity; (**B**) fluorescence decay curves of fluorescein induced by AAPH in the presence of Trolox and PBSR1 with time. The results were compared with the positive control (V_c_).

**Figure 11 foods-12-01758-f011:**
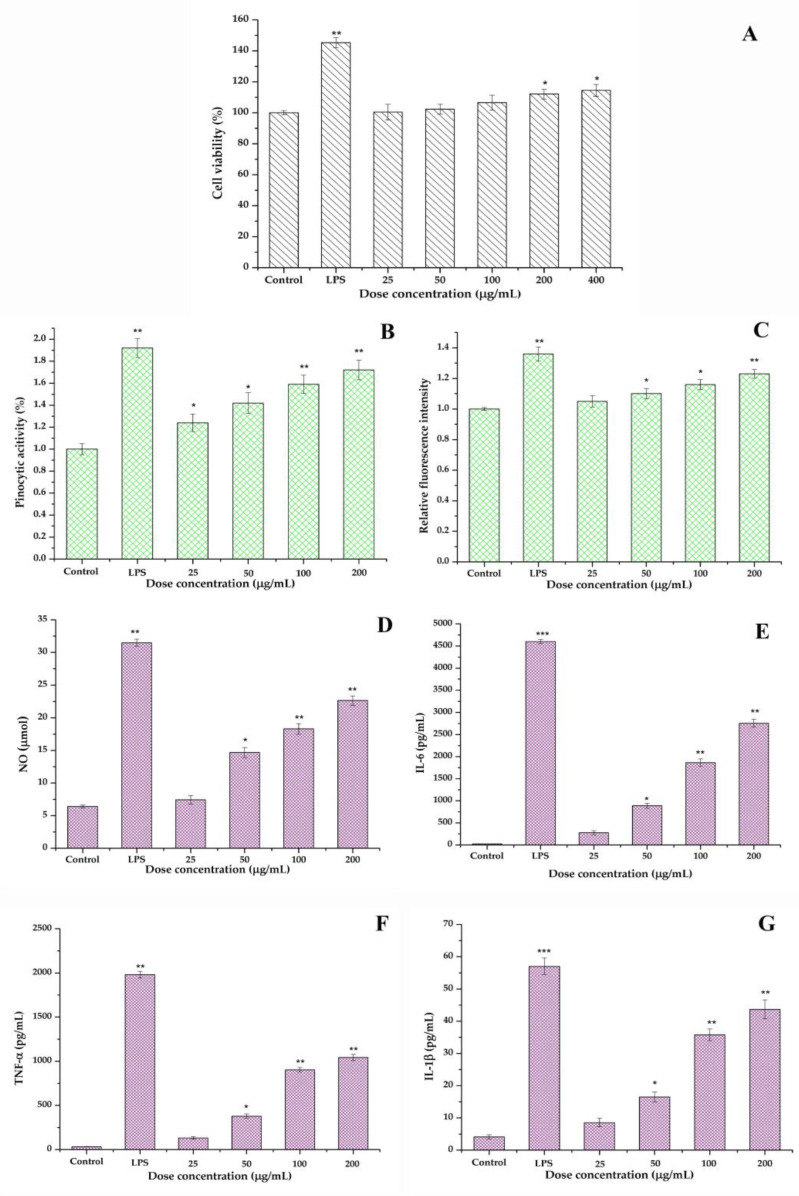
Effects of PBSR1 treatment on the behaviors of RAW264.7 cells and cytokines secretion: cell viability (**A**), pinocytosis rate of neutral red (**B**), ROS production (**C**), NO secreted level (**D**), IL-6 secreted level (**E**), TNF-α secreted level (**F**) and IL-1β secreted level (**G**). Note: each experiment was tested in triplicate, and the error bars are standard deviations and significant differences of cell viability, pinocytosis rate, ROS production and different cytokines release of samples vs. control group are presented by * *p* < 0.05, ** *p* < 0.01 and *** *p* < 0.001.

**Table 1 foods-12-01758-t001:** Methylation analysis of PBSR1 based on GC-MS results.

Residues	Retention Time (min)	PMAAs	Type of Linkages	Molar Ratio (mol%)	Major Mass Fragments (*m/z*)
1	36.52	2,4,6-Me_3_-Gal*p*	1,3-linked Gal*p*	14.3	87, 101, 118, 129, 143, 161, 174, 202, 217, 277
2	37.89	2,3,4-Me_3_-Gal*p*	1,6-linked Gal*p*	85.7	87, 99, 102, 118, 129, 145, 162, 189, 233

**Table 2 foods-12-01758-t002:** Chemical shifts of PBSR1 (δ, ppm).

Residue	Chemical Shifts (δ, ppm)					
	H1/C1	H2/C2	H3/C3	H4/C4	H5/C5	H6/C6
(A) β-1,6-D-Gal*p*	4.46/103.31	3.55/70.73	3.66/72.64	3.93/68.92	3.73/75.1	3.93, 4.06/69.01
(B) β-1,3-D-Gal*p*	4.71/103.78	3.78/70.06	3.89/81.38	4.13/68.52	3.95/76.55	3.79/60.91

## Data Availability

Data are contained within the article.

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
