# Peer review of "Molecular Characterization and Bioactivities of a Novel Polysaccharide from Phyllostachys pracecox Bamboo Shoot Residues"

_foods, 2023, doi:10.3390/foods12091758_

Round 1

Reviewer 1 Report

This article is devoted to the isolation and characterization of a new polysaccharide from Phyllostachys pracecox bamboo shoot residues. The article is relevant because polysaccharides have a wide potential for practical use - both as an independent biologically active agent and as a platform molecule for further chemical modification. This article provides a sufficient amount of experimental data that complement each other. In general, the article makes a good impression. However, there are some points that it is desirable to correct:

1. It is desirable to compare the arabinogalactan obtained by the authors from Phyllostachys pracecox in more detail with arabinogalactans from other raw materials (for example, Siberian larch). At this point, it is desirable to pay attention to the molecular and other physico-chemical characteristics of the polysaccharide.

2. Please more clearly indicate the ratio of arabinose to galactose in the resulting polysaccharide.

3. Why did the authors choose this particular method for obtaining arabinogalactan? The literature describes various methods for obtaining arabinogalactans from plant materials.

4. Often, along with the release of polysaccharides (in particular, galactans), flavonoids are also released. How do authors define product purity?

5. A clearer relationship between all methods of analysis is needed. One method should complement the other. They don't have to be separate. This is a holistic study and the description must also be holistic.

6. It is desirable to add more literature comparison when describing the resulting polysaccharide.

7. Please cite: 10.1007/s13399-021-02250-x.

8. It is desirable to bring more clarity to the prospects for the use of the polysaccharide obtained by the authors.

Author Response

 Response to the Reviewers’ Comments

Dear Editor,

Thank you very much for your letter and the reviewers’ reports. Based on your comment and request, we have carefully proof-read and made extensive modification on the original manuscript.

List of Actions:

1: The revised paragraphs are in red type.

2: The new figure and some supplements are revised in red type.

3: Response to reviewer’s comments are appended below.

Here, we attach revised manuscript for your approval. A document answering every question from the editor is also summarized and enclosed, and our description on revision according to the reviewers’ comments is appended below. We appreciate for editor’s warm work earnestly, and hope that the corrections will meet with approval.

Looking forward to hearing from you.

Sincerely yours,

Liang He

*****************************************************

Reviewer 1: This article is devoted to the isolation and characterization of a new polysaccharide from Phyllostachys pracecox bamboo shoot residues. The article is relevant because polysaccharides have a wide potential for practical use - both as an independent biologically active agent and as a platform molecule for further chemical modification. This article provides a sufficient amount of experimental data that complement each other. In general, the article makes a good impression. However, there are some points that it is desirable to correct:

Question 1: It is desirable to compare the arabinogalactan obtained by the authors from Phyllostachys pracecox in more detail with arabinogalactans from other raw materials (for example, Siberian larch). At this point, it is desirable to pay attention to the molecular and other physico-chemical characteristics of the polysaccharide.

Comment: Thank you so much for your valuable suggestion. We have already made more comparison of other reported arabinogalactans in the related parts of Results in the manuscript. For instance, compared with other arabinogalactans, PBSR1 significantly differs from other Siberian larch AG (a weight average molecular mass of 9000-13,000) by its relatively higher molecular mass. Even though, its PDI was much lower than this kind of Siberian larch AG (PDI:1.9-2.3).

Question 2: Please more clearly indicate the ratio of arabinose to galactose in the resulting polysaccharide.

Comment: Thanks for your valuable suggestion. We have already added the missing information of Gal/Ara ratio in the secession of monosaccharide compositions of PBSR1 and some more discussion was provided as well.

Question 3: Why did the authors choose this particular method for obtaining arabinogalactan? The literature describes various methods for obtaining arabinogalactans from plant materials.

Comment: Thanks for your suggestion. Normally, the AG mass fraction is from 15% to 30% as an important hemicellulose existed in many larch species. So it can be easily acquired by many simple techniques such as membrane filtration, ultracentrifugation and other macroporous resin adsorption. However, the polysaccharide (PBSR1) in bamboo shoots or other natural resources like sapwoods, only occupy less than 1% of total fractions. It is hard to get a certain amount of the bioactive polymers by those methods. Tremendous studies have shown that natural polysaccharides can be purified by using ion exchange column and gel chromatography. Moreover, some carbohydrate compounds as second metabolites exist in side bamboo shoots, which may have charge differences with each other. Overall, we adopted the chromatography strategy including ion exchange and gel permeation to get the purified arabinogalactan from bamboo shoots residues.

Question 4: Often, along with the release of polysaccharides (in particular, galactans), flavonoids are also released. How do authors define product purity?

Comment: Thanks for your reminder. The flavonoids contents in the dry PBSR1 were determined by the measurement of absorbance intensity at 400 nm caused by the flavonoid complexes with aluminum chloride. In this study, flavonoid content was 1.67% in the dry extract, while carbohydrate content occupied almost 97.38% in the dry sample. That information has been added in the related parts.

Question 5: A clearer relationship between all methods of analysis is needed. One method should complement the other. They don't have to be separate. This is a holistic study and the description must also be holistic.

Comment: Thanks for your suggestion. We have already provided more information on the relationship between those methods and their mutual corroboration in the related parts.

Question 6: It is desirable to add more literature comparison when describing the resulting polysaccharide.

Comment: Thanks for your suggestion. We have already made more comparison of PBSR1 with other reported arabinogalactans, especially some AG-II from larch species when explaining the physicochemical properties of PBSR1 and its unique structure characteristics.

Question 7: Please cite: 10.1007/s13399-021-02250-x.

Comment: Thanks for your suggestion. We have already cited this article in the Results section of 3.3. Homogeneity and Molecular Weight of PBSR1 and made more discussion.

Question 8: It is desirable to bring more clarity to the prospects for the use of the polysaccharide obtained by the authors.

Comment: Thanks for your suggestion. Bamboo shoot residues remain as agro-industrial wastes causing serious environmental and disposal problem. However, discarded PBSR are still good sources of dietary polysaccharides. Converting PBSR into products with added economic value is of great significance and worth further study. Our study demonstrated that a novel polysaccharide isolated from PBSR had potent antioxidant and immunomodulatory activities. Moreover, the deep investigation of relationship between structure and function provided us some important information that its moderate Mw, β-(1→6)-galactopyranosyl backbone and triple-helix conformation may be beneficial to its strong biological activities. All those findings boosted PBSR1 to be developed as food-additives or healthy food in functional food and pharmaceutical industries. The related information has been added in the main parts.

Once again, special thanks for your comments, and we are looking forward to learning much more from you.

Yours sincerely

Liang He

Reviewer 2 Report

Manuscript ID: foods-2270303, titled “Structural characterization and biological activities of a novel polysaccharide from Phyllostachys pracecox bamboo shoot residues” has great importance in the field of phytochemistry and pharmaceutical applications. But there are some comments:

New isolated polysaccharide was mentioned through the text on several places with two abbreviations (PBSR and PBSR1); be kindly asked to uniform that.

In the Abstract (line 18) authors said chromatograph instead of chromatography; please correct it.

In the Introduction, lines 36-38 and 66-68 please be advised to check the sense of the sentences. In general, introduction part should be more oriented to agro-industrial waste along with its potential exploiting in manufacturing of added-value products. The last paragraph of the Introduction was dedicated to description of all measured parameters what is not usual for background, in fact it should state relevance of the study design with solid hypothesis.

Moving to M&M part, extraction and fractionation of crude polysaccharide include centrifugation on 8000 rpm, here is missing information of the centrifuge diameter or it would be better to precise rcf units? Next methodological flaw is that the DPPH assay was performed on three samples (line 186); which; it was remained unclear in the results? Finally, you applied statistical analysis only in immunomodulatory measurements; what happened with antioxidant activity?

In the Results (line 463) DPPH scavenging ability of PBSR1 was compared to Vc (is it vehicle?), please explain on both places, text and figure. All Figures have to have legend in details so that they can be followed as separate parts of the manuscript.

Discussion part is too short. It would be fine to make it more comprehensive since this study assessed the PBSR1 structural and biological characterization and their relationship. The authors discussed better antioxidant activity of PBSR1 without specifying in relation to what (lines 585-586).

Author Response

Response to the Reviewers’ Comments

Dear Editor,

Thank you very much for your letter and the reviewers’ reports. Based on your comment and request, we have carefully proof-read and made extensive modification on the original manuscript.

List of Actions:

1: The revised paragraphs are in red type.

2: The new figure and some supplements are revised in red type.

3: Response to reviewer’s comments are appended below.

Here, we attach revised manuscript for your approval. A document answering every question from the editor is also summarized and enclosed, and our description on revision according to the reviewers’ comments is appended below. We appreciate for editor’s warm work earnestly, and hope that the corrections will meet with approval.

Looking forward to hearing from you.

Sincerely yours,

Liang He

*****************************************************

Reviewer 2: Manuscript ID: foods-2270303, titled “Structural characterization and biological activities of a novel polysaccharide from Phyllostachys pracecox bamboo shoot residues” has great importance in the field of phytochemistry and pharmaceutical applications. But there are some comments:

Question 1: New isolated polysaccharide was mentioned through the text on several places with two abbreviations (PBSR and PBSR1); be kindly asked to uniform that.

Comment: Thank you so much for your reminder. We have checked all the places with these two abbreviations and made them uniform. As a matter of fact, PBSR means the crude polysaccharides extracted from Phyllostachys pracecox bamboo shoot residues. While PBSR1 refers to the first fraction in crude polysaccharide after isolated by DEAE-Sepharose FF column.

Question 2: In the Abstract (line 18) authors said chromatograph instead of chromatography; please correct it.

Comment: Thanks for your suggestion. We have already revised that word into chromatography.

Question 3: In the Introduction, lines 36-38 and 66-68 please be advised to check the sense of the sentences. In general, introduction part should be more oriented to agro-industrial waste along with its potential exploiting in manufacturing of added-value products. The last paragraph of the Introduction was dedicated to description of all measured parameters what is not usual for background, in fact it should state relevance of the study design with solid hypothesis.

Comment: Thanks for your valuable suggestion. We have already corrected the sentences in these two places. Moreover, we have modified the last paragraph of introduction into potential exploitation of this possible high-valued agro-industrial wastes and stressed the final purpose of our study.

Question 4: Moving to M&M part, extraction and fractionation of crude polysaccharide include centrifugation on 8000 rpm, here is missing information of the centrifuge diameter or it would be better to precise rcf units? Next methodological flaw is that the DPPH assay was performed on three samples (line 186); which; it was remained unclear in the results? Finally, you applied statistical analysis only in immunomodulatory measurements; what happened with antioxidant activity?

Comment: Thanks for your valuable suggestion. We have provided the relative centrifugal force of 10600×g in the related method. Further we have corrected the DPPH assay method for PBSR1 and made a clear explanation of statistical analysis in antioxidant measurements as well.

Question 5: In the Results (line 463) DPPH scavenging ability of PBSR1 was compared to Vc (is it vehicle?), please explain on both places, text and figure. All Figures have to have legend in details so that they can be followed as separate parts of the manuscript.

Comment: Thanks for your suggestion. We have already explained that both in the related place and figure 10. Moreover, we have provided all the figure legends in details under each figure for better understanding.

Question 6: Discussion part is too short. It would be fine to make it more comprehensive since this study assessed the PBSR1 structural and biological characterization and their relationship. The authors discussed better antioxidant activity of PBSR1 without specifying in relation to what (lines 585-586).

Comment: Thanks for your suggestion. We have already provided more profound discussion and emphasized the relationship of structure-function based on the obtained results in the section of Discussion. Moreover, we also added the missing terms of “compared to other polysaccharides from Chimonobambusa quadrangularis processing by-products” in the parts of antioxidant activity of PBSR1.

Once again, special thanks for your comments, and we are looking forward to learning much more from you.

Yours sincerely

Liang He

Reviewer 3 Report

This manuscript contains a careful and well designed article, presenting a new polysaccharide compound derived from plant food byproducts. Some of its interesting and, quite likely, useful properties are studied and described here. Although the study follows the lines of previous works by the same authors on other similar compounds, I believe the newly described one is worth to be reported, and readers can find both methods and results of interest.

The Methods section is correctly presented and contributes to the clarity and value of the article. Use of English is generally acceptable, although it should be improved in different parts of the manuscript, especially in the first paragraph of the Introduction. Some too enthusiastic descriptions, such as “famous”, “the best” or “warmly”, should be avoided.

Figure 1 contains some mistakes in lettering. NaCl must have the “C” capitalized. Besides, some of the text has too small size to be easily read. Figure 2 and 11 have inconsistent fonts and in some cases they are deformed, flattened. Figure 3 has also inconsistent and deformed fonts and the insert is actually illegible, even after expanding. Figure 7, 8 and 10 also contain too small size text (especially the insert of Fig. 10).

The sentence “Moreover, 1.33 of the polydispersity reflected a relatively uniformed distribution” (line 280), is not correct, please correct.

In line 407: “An overwhelming” something is missing here: number? percentage?... “of those tended to be aggregated”.

What is meant by “elaborated” in: “Our AFM observation further elaborated the analysis of NMR and light scattering” (line 416).

About AFM findings (lines. 404-411), I wonder if the association of different chains may be mostly meaningless, as an artifact, resulting from aggregation on the drying process. Maybe some words should be included on this in the text.

“…expectively…” (line. 427), what is meant?

In lines 485-486, I do not understand the sentence: “The higher purified PBSR1 with streched chain molecules could easily provide hydrogen to stabilize free radicals or directly react with them to terminate the chemcial reaction”.

Otherwise, the article has no problem for its publication, as far as I am concerned.

I would advise choosing different keywords: they are used to add information to that contained in the title, when readers make a search. If you use words not contained in the title, more searches can lead to your article.

Author Response

Response to the Reviewers’ Comments

Dear Editor,

Thank you very much for your letter and the reviewers’ reports. Based on your comment and request, we have carefully proof-read and made extensive modification on the original manuscript.

List of Actions:

1: The revised paragraphs are in red type.

2: The new figure and some supplements are revised in red type.

3: Response to reviewer’s comments are appended below.

Here, we attach revised manuscript for your approval. A document answering every question from the editor is also summarized and enclosed, and our description on revision according to the reviewers’ comments is appended below. We appreciate for editor’s warm work earnestly, and hope that the corrections will meet with approval.

Looking forward to hearing from you.

Sincerely yours,

Liang He

*****************************************************

Reviewer 3: This manuscript contains a careful and well designed article, presenting a new polysaccharide compound derived from plant food byproducts. Some of its interesting and, quite likely, useful properties are studied and described here. Although the study follows the lines of previous works by the same authors on other similar compounds, I believe the newly described one is worth to be reported, and readers can find both methods and results of interest.

Question 1: The Methods section is correctly presented and contributes to the clarity and value of the article. Use of English is generally acceptable, although it should be improved in different parts of the manuscript, especially in the first paragraph of the Introduction. Some too enthusiastic descriptions, such as “famous”, “the best” or “warmly”, should be avoided.

Comment: Thank you so much for your suggestion. We have already deleted all those improper descriptions in the paper, especially mentioned words of “famous”, “the best” and “warmly”. And the English in whole manuscript has also been improved by a native speaker.

Question 2: Figure 1 contains some mistakes in lettering. NaCl must have the “C” capitalized. Besides, some of the text has too small size to be easily read. Figure 2 and 11 have inconsistent fonts and in some cases they are deformed, flattened. Figure 3 has also inconsistent and deformed fonts and the insert is actually illegible, even after expanding. Figure 7, 8 and 10 also contain too small size text (especially the insert of Fig. 10).

Comment: Thanks for your suggestion. We have already revised carefully all the figures to make the fonts uniformed and the revised figures have been presented more clearer and readable for readers.

Question 3: The sentence “Moreover, 1.33 of the polydispersity reflected a relatively uniformed distribution” (line 280), is not correct, please correct.

Comment: Thanks for your valuable suggestion. We have already corrected this sentence in the text.

Question 4: In line 407: “An overwhelming” something is missing here: number? percentage?... “of those tended to be aggregated”.

Comment: Thanks for your valuable suggestion. We have already added the missing word “percentage” in the related part.

Question 5: What is meant by “elaborated” in: “Our AFM observation further elaborated the analysis of NMR and light scattering” (line 416).

Comment: Thanks for your suggestion. We have already replaced ““elaborated” with “proved” in the part.

Question 6: About AFM findings (lines. 404-411), I wonder if the association of different chains may be mostly meaningless, as an artifact, resulting from aggregation on the drying process. Maybe some words should be included on this in the text.

Comment: Thanks for your suggestion. We have already added the explanation of those aggregation caused by drying process.

Question 7: “…expectively…” (line. 427), what is meant?

Comment: Thanks for your reminder. We have already deleted this word in the text.

Question 8: In lines 485-486, I do not understand the sentence: “The higher purified PBSR1 with streched chain molecules could easily provide hydrogen to stabilize free radicals or directly react with them to terminate the chemcial reaction”.

Comment: Thanks for your suggestion. We have already revised this sentence to be more clarified: “The stretched PBSR1 molecules might make hydroxyl groups on the chain to be exposed to the environment, easily providing hydrogen to stabilize free radicals or directly react with them to terminate the chemical reaction”.

Question 9: I would advise choosing different keywords: they are used to add information to that contained in the title, when readers make a search. If you use words not contained in the title, more searches can lead to your article.

Comment: Thanks for your suggestion. We have already revised some keywords in order to enlarge the scope search for readers.

Once again, special thanks for your comments, and we are looking forward to learning much more from you.

Yours sincerely

Liang He

Round 2

Reviewer 1 Report

accepted

Author Response

Thank you so much for the whole review process and valuable suggestions

Reviewer 2 Report

The authors tried to respect almost all my comments and suggestions. They worked on improving the Introduction part as they modified the last paragraph highlighting potential exploitation of this high-valued agro-industrial wastes and stressed the final purpose of their study. It is still desirable to expand discussion part after what the manuscript could be accepted.

Author Response

 Response to the Reviewers’ Comments

Dear Editor,

Thank you very much for your letter and the reviewers’ reports. Based on your comment and request, we have carefully proof-read and made extensive modification on the revised manuscript.

List of Actions:

1: The second revised paragraphs are in red type with yellow highlight.

2: Response to reviewer’s comments are appended below.

Here, we attach revised manuscript for your approval. A document answering every question from the editor is also summarized and enclosed, and our description on revision according to the reviewers’ comments is appended below. We appreciate for editor’s warm work earnestly, and hope that the corrections will meet with approval.

Looking forward to hearing from you.

Sincerely yours,

Liang He

*****************************************************

Reviewer 2:

Question 1: The authors tried to respect almost all my comments and suggestions. They worked on improving the Introduction part as they modified the last paragraph highlighting potential exploitation of this high-valued agro-industrial wastes and stressed the final purpose of their study. It is still desirable to expand discussion part after what the manuscript could be accepted.

Comment: Thank you so much for your suggestion. We have already made more discussion in the parts of results, especially in the deep investigation of structure-function relationship of PBSR1 after the section of immunomodulatory activities with yellow highlights in the manuscript.

Academic editor:

Question 1: Please improve the quality of Figure 7 and 11.

Comment: Thank you for the suggestion. We have improved the quality of both figure7 and figure 11 significantly and the new figures have been replaced in the paper.

Once again, special thanks for your comments, and we are looking forward to learning much more from you.

Yours sincerely

Liang He
